# Water-Soluble Vitamins and Trace Elements Losses during On-Line Hemodiafiltration

**DOI:** 10.3390/nu14173454

**Published:** 2022-08-23

**Authors:** Alban Bévier, Etienne Novel-Catin, Emilie Blond, Solenne Pelletier, Francois Parant, Laetitia Koppe, Denis Fouque

**Affiliations:** 1Department of Nephrology, Dialysis and Nutrition, Hospices Civils de Lyon, Lyon Sud Hospital, 69495 Pierre Bénite, France; 2Department of Biochemistry, Hospices Civils de Lyon, Lyon Sud Hospital, 69495 Pierre Bénite, France; 3INSERM U1060, CarMeN Laboratory, Lyon 1 Claude Bernard University, INRAE U1397, 69495 Pierre Bénite, France

**Keywords:** water-soluble vitamins, trace elements, hemodiafiltration, dialysate, loss, mass transfer

## Abstract

Maintenance hemodialysis induces water-soluble vitamins and trace elements losses, which is why recommendations regarding potential supplementation were provided, but mainly based on conventional hemodialysis. This study′s aim was to measure the water–soluble vitamins and trace element losses during one on-line post-dilution hemodiafiltration (HDF) session. Thirty-nine patients under maintenance HDF were enrolled. We used the Theraflux^®^ sampler (Theradial Corp., Orvault, France) to analyze the full session dialysate mass transfer. Blood and dialysate samples were collected before and after one HDF session to measure B1, B2, B6, B9, B12, C vitamins, zinc, and selenium concentrations. Values significantly decreased for B1 (20.2%), B2 (13%), B6 (25.4%), B9 (32.6%), C (66.6%) and selenium (6.7%). No significant differences were found for vitamin B12 and zinc. The dialysate losses per session were 1.12 ± 0.88 mg for vitamin B1, 0.28 ± 0.30 mg for B2, 0.33 ± 0.09 mg for B6, 0.3 ± 0.18 mg for B9, 147.5 ± 145.50 mg for C and 25.75 ± 6.91 mg for zinc. Vitamin B12 and selenium were under detection values. In conclusion, during a standard 4hr-HDF session, we found important losses for vitamin B1, B6, B9, C and zinc, suggesting the need for regular monitoring of plasma levels and systematic supplementation of these compounds.

## 1. Introduction

Hemodialysis is one of the available techniques for treatment of chronic renal failure by restoring extracellular and intracellular fluid homeostasis, thanks to exchange between blood and dialysate through a semi-permeable membrane.

On maintenance hemodialysis (MHD), patients are at risk of water–soluble vitamins and trace elements deficiency [1,2]. The origin of these deficiencies is not fully understood and involves multiple mechanisms such as decreased food intake, gastrointestinal malabsorption, altered vitamin metabolism and loss during the hemodialysis session itself [2,3].

These deficiencies may lead to pathological conditions, such as scurvy, encephalopathy, anemia, hyperoxaluria, oxidant stress, and endothelial dysfunction. Consequently, clinical practice guidelines recommend systematic supplementation of certain compounds for hemodialysis (HD) patients [4,5,6]. Among MHD, patients′ prescriptions of water-soluble vitamins and trace elements are nowadays more prevalent than before these recommendations were established. However, these recommendations are based on conventional hemodialysis studies, and might not be appropriate to on-line post-dilution hemodiafiltration (HDF), a relatively new modality of dialysis that is now widely available in European HD centers [7].

On-line HDF indeed associates diffusive clearance with convective clearance using high-flux membranes to achieve better removal of middle molecules, some of which are also considered as uremic toxins. This process seems to improve the morbi-mortality of maintenance hemodialysis patients if a large convective volume is reached [8,9].

It is unknown if the loss of water-soluble vitamins and trace elements is affected by HDF compared to conventional hemodialysis. Therefore, we conducted a prospective study to measure the water-soluble vitamins and trace elements losses during one on-line post-dilution HDF treatment.

## 2. Materials and Methods

### 2.1. Patients and Study Design

From July 2021 to December 2021, 39 patients from Lyon Sud Hospital hemodialysis center were enrolled in this prospective study. All were maintenance HD patients treated by on-line post–dilution HDF using the same high flux poly-sulfone dialyzer (FX1000 dialyzer, Fresenius^®^, Bad Homburg vor der Höhe, Germany).

The inclusion criteria were adult patients with a dialysis vintage of at least three months, undergoing 4 h sessions of on–line HDF thrice weekly, with a vascular access allowing a blood flow greater than 300 mL/min. The exclusion criteria included having a history of an active infectious, inflammatory or malignant disease, or intestinal malabsorption. Patients were also excluded if they presented with hypersensitivity to poly-sulfone dialyzers.

In our unit, patients received a systematic weekly supplementation of vitamins and trace elements composed of 1 thiamine tablet of 250 mg, 1 pyridoxin tablet of 250 mg, 1 folate tablet of 5 mg, and 1 ascorbic acid tablet of 500 mg. By contrast, vitamin B12, zinc and selenium were only supplemented after semestrial plasma level monitoring and values below normal.

### 2.2. Blood Sampling

Vitamins and trace element blood levels were measured during the semestrial work up with one sample taken at the beginning and one sample at the end of the session from the ‘arterial line’ of the extracorporeal circuit.

### 2.3. Dialysate Sampling

The dialysate was collected thanks to a novel sampling device (THERAFLUX^®^, Theradial Corp., Orvault, France) which samples 1% of the total effluent volume during the hemodialysis session. This allowed an accurate and more convenient approach compared to former devices that collected the totality of dialysate (which can represent a volume of up to 200 L for one single four-hour HDF session).

The dialysate was collected, sheltered from light and put on ice to prevent degradation of the collected nutrients. At the end of the session, dialysate samples were immediately addressed to our biochemistry laboratory for technical processing. The volume of the collected effluent sample and total effluent volume during the HDF session were provided by the device.

### 2.4. Biochemical Measurements

For thiamine pyrophosphate (TPP) and pyridoxal phosphate (PP) measurements, blood samples were collected in Tripotassium EDTA VacutainerTM blood collection tubes (Becton Dickinson BD UK Limited, Oxford, England). Briefly, aliquots of whole blood were directly frozen at <−18 °C before analysis. TPP and PP were then extracted and derived after the use of high-pressure liquid chromatography Chromsystems reagent kits (Chromsystems Instruments & Chemicals GmbH, München, Germany) to precipitate proteins by acidification, to neutralize supernatants and to derivate them with specific fluorophore to allow fluori-metric detection of these compounds. Supernatants containing derived TPP and PP were then injected in a Ultra-High Performance Liquid Chromatography (UHPLC) system coupled to fluori-metric detection (H-Class systems, Waters, St Quentin-en-Yvelines, France). Quantification was performed by using Empower3_HF1_Enterprise software (version 7.30.00.00, Waters). Mobile phase and chromatographic columns were provided by Chromsystems laboratory. Wavelengths used for TPP determination were 367 nm for excitation wavelength and 435 nm for emission wavelength, whereas wavelengths used for PP determination were 320 nm for excitation wavelength and 415 nm for emission wavelength. For TPP, PLP, thiamine (T) and pyridoxal (P) quantification in dialysate, aliquots were directly frozen at <−18 °C before analysis. TPP, PLP, T and P were then extracted after protein precipitation in acidified condition (10% *w*/*v* of trichloroacetic solution) and supernatant was injected via a liquid chromatography tandem mass spectrometry (LCMSMS) system (Ultimate 3000/TSQ Quantiva system (Thermo Fisher Scientific, Courtaboeuf, France)) to allow quantification. Quantification was performed by using TraceFinder software (Thermo Fisher Scientific). Separation was achieved on a Symmetry C18 column. A gradient elution utilizing 0.1% formic acid (FA) in water as solvent A and 0.1% FA in methanol as solvent B was performed having a varying flow rate, a nonlinear gradients steps and a total run of 3 min. The gradient was as follows: 0 min (100% A and 0% B, 0.3 mL/min, 0.6 min (90% A and 10% B, 0.6 mL/min), 1.9 min (3% A and 97% B, 0.6 mL/min) and 2.4 min (100% A and 0% B, 0.3 mL/min). TPP, PLP, T and P were measured by electrospray ionization in positive ionization mode with the following selected mass transitions: *m*/*z* 425.1 > 121.8 and 425.1 > 303.9 for TPP, *m*/*z* 428.2 > 124.9 for TPP-d3, *m*/*z* 265.0 > 122.1 and *m*/*z* 265.0 > 144.1 for T, *m*/*z* 248.1 > 150.0 and 248.1 > 94.0 for PLP, *m*/*z* 251.0 > 153.1 for PLP-d3, *m*/*z* 168.0 > 150.0 and 168.0 > 94.1 for P. Other mass spectrometer parameters were capillary voltage 3.0 kV, sheath gas 40, auxiliary gas 10, sweep gas 1, and desolvation temperature 350 °C. Argon was used as the collision gas.

For riboflavin, flavin adenine dinucleotide (FAD), and flavin mononucleotide (FMN) measurements, blood samples were collected in Tripotassium EDTA VacutainerTM blood collection tubes (Becton Dickinson BD UK Limited, Oxford, England). Briefly, aliquots of whole blood were directly frozen at <−18 °C before FAD, FMN and riboflavin analysis. FAD, FMN and riboflavin were then extracted after protein precipitation using Chromsystems reagent kits (Chromsystems Instruments & Chemicals GmbH, Gräfelfing, Germany). Supernatants containing FAD, FMN and riboflavin were then injected in an isocratic UHPLC system coupled to fluori-metric detection (Waters, St Quentin-en-Yvelines, France). Quantification was performed by using Empower3_HF1_Enterprise software (version 7.30.00.00, Waters). Mobile phase and chromatographic columns were provided by Chromsystems laboratory. Analysis of FAD, FMN and riboflavin used a specific excitation wavelength of 465 nm and a specific emission wavelength of 525 nm, allowing specific determination of these three components. Due to their native fluorescence, riboflavin and its actives metabolites, FMN and FAD, could be detected directly without derivatization. For dialysate measurements, aliquots were directly frozen at <−18 °C before riboflavin analysis. Then, the same method as for blood was used to quantify riboflavin.

Blood vitamin C concentration was measured by high performance liquid chromatography coupled to electrochemical detection with a H-Class Waters chromatographic system (Waters, St Quentin-en-Yvelines, France) and Empower3_HF1_Enterprise software (version 7.30.00.00, Waters). Briefly, to guarantee stability of vitamin C, blood samples collected on lithium heparin tube were centrifuged at 2500× *g* during 10 min for a maximum of 3 h following blood collection. Supernatant was stabilized in a solution containing 6% of sulfosalicylic acid to induce precipitation of proteins, EDTA (2 mM) to chelate bivalent ions, which could disturb electrochemical measurement, and *N*-ethylmaleimide (20 mM) to stabilize vitamin C in its reduced form. After acidification and stabilization, diluted sample was frozen at <−18 °C and analyzed the following week. We demonstrated, in a previous study, that vitamin C concentration was stable in acidified and stabilized plasma conserved at <−18 °C for one month [10]. For Vitamin C measurement, aliquots of vitamin C, previously stabilized and frozen, were centrifuged at 15,000× *g* for 10 min. Then, the supernatant was diluted in mobile phase and injected via a chromatographic system. Separation was accomplished using isocratic elution with high performance liquid chromatography mobile phase vitamin C (Chromsystems, Gräfelfing, Germany) containing 3 M of KCl (Merck KGaA, Darmstadt, Germany) on a reverse column for vitamin C measurement (Chromsystems, Gräfelfing, Germany) at 37 °C. Ascorbic acid was oxidized at a potential of +0.60 V and electrons generated during this oxidation reaction were quantified. With this method, we only measured vitamin C in its reduced form. No internal standard was used. Neither liquid/liquid nor solid/liquid extractions of vitamin C were performed. Plasma for vitamin C determination was only diluted in acidification/stabilization solution and in the mobile phase. Dialysate, to guarantee its stability, it was centrifuged at 2500 g for 10 min at a maximum of 3 h following its collection. Vitamin C quantification was the same in dialysate as in blood.

For vitamins B9 and B12, blood samples were collected in a VacutainerTM tube with clot activator (silica) and separating gel (Becton Dickinson BD UK Limited, Oxford, England). Vitamins B9 and B12 concentrations were measured in serum by Chemiluminescent Microparticle Immunoassay (CMIA) technology on the architect system (Abbott, France). The architect vitamin B9 and vitamin B12 assay is a two-step assay for quantitative determination in serum. In the first step, samples and pre-treatment reagent of architect folate or vitamin B12 kit are combined. An aliquot of the pre-treated sample, assay diluent, and intrinsic factor coated paramagnetic microparticles for vitamin B12 measurement or Folate Binding Protein (FBP) coated paramagnetic microparticles for vitamin B9 measurement are combined. Vitamins B12 and B9 binds respectively to the intrinsic factor or FBP coated microparticles. After washing, vitamins B12 or B9 acridinium labelled conjugate is added in this second step. Pre-Trigger and Trigger Solutions are then added to the reaction mixture; the resulting chemiluminescent reaction is measured as relative light units (RLUs). An inverse relationship exists between the amount of B12 or B9 in the sample and the RLUs detected by the ARCHITECT i optical system. Dialysate concentrations were measured with the same method as for blood.

For trace elements, blood samples were collected from the patients into 10 mL capacity BD Vacutainer^®^ K2E (EDTA) plastic tube. Levels of Selenium (Se) were determined by Inductively Coupled Plasma–Mass Spectrometry (ICP–MS) on a NexION 350X instrument (PerkinElmer) in Dynamic Reaction Cell (DRC) mode using oxygen as the reaction gas. The internal standard was rhodium. Levels of Zinc (Zn) were determined by Inductively Coupled Plasma–Optical Emission Spectrometry (ICP–OES) on a 5110 instrument (Agilent). The emission wavelength was 213.857 nm. The laboratory participates in external proficiency testing programs for trace elements. Dialysate concentrations were measured with the same method as for blood.

Because hemoconcentration occurs during the HD session, post dialysis concentrations were corrected using the relative Blood Volume Measurement (rBVM) value [11], provided by the generator software (Fresenius 5008^®^). We used the following equation: [Nutrients]Corrected = ([rBVM]post–HD/[rBVM]pre–HD) x [Nutrients]post–HD

with [rBVM]post–HD, rBVM at the end of the session;

[rBVM]pre–HD, rBVM at the beginning of the session;

and [Nutrients]post–HD, blood nutrients concentrations after the hemodialysis session.

### 2.5. Dialysis Parameters

The convective volume was calculated as the addition of ultrafiltration volume (total volume removed from patient during the dialysis session) with the substitution volume (post–dilution reinjected volume). Kt/V was provided by the dialysis machine using ionic dialysis. The blood flow corresponded to the average blood flow during the dialysis session. R–BVM was recorded at the end of the dialysis session to reflect hemoconcentration.

### 2.6. Statistical Analysis

Data were analyzed using Graphpad Prism 5.0 (GraphPad software, La Jolla, CA, USA) software. Results are expressed as mean ± standard deviation (SD). To compare blood concentration before and after the HDF session, we used a paired *t*–test with two tailed *p* values. A *p* value < 0.05 was considered statistically significant in all analysis.

For analyses involving the dialysate, we used the mean concentration ± SD and compared the amount of each nutrient loss with the recommended dietary allowance (RDA, defined as the average daily dietary intake of an essential nutrient that is sufficient to meet the requirement for that nutrient in nearly all (97–98%) of healthy individuals of a given life stage and gender). To determine the total amount of nutrients lost during one session, we multiplied the concentration of each compound found in the dialysate sampled by the total effluent volume.

This research was approved by the local institutional review committee (Comité de Protection des Personnes–Recherche Biomédicale, CPP Lyon Sud–Est IV) and conducted in accordance with its ethical standards and the principles expressed in the Declaration of Helsinki. All subjects involved in the research were informed and had given their non–opposition prior to enrolment. The Clinical Trial identifier of this study was NCT05099185.

## 3. Results

### 3.1. Baseline Characteristics

All patient characteristics are listed in Table 1. Patients were in maintenance hemodialysis, with a dialysis vintage of 44 ± 33 months, aged 67.0 ± 14.8 years with a mean BMI of 27.6 ± 5.7 and a proportion of 25% female. The mean Kt/V was 1.56 ± 0.23 and the mean convective volume 25.5 ± 2.5 L. One hemodialysis session consumed an average of 166.2 ± 29.6 L of dialysate. Patients were well nourished (mean albumin 38.2 ± 3.8 g/L, mean pre–albumin 11.0 ± 0.08 g/L).

### 3.2. Blood Concentrations of Vitamins and Trace Elements

The different results of blood concentration are disclosed in Table 2, showing the normal range, the pre- and post–session concentration, and percentage of blood decrease during the dialysis session. Individual variations of each compound are shown in Figure 1, Figure 2, Figure 3, Figure 4, Figure 5, Figure 6, Figure 7 and Figure 8.

Mean whole blood vitamin B1 (thiamine pyrophosphate), B2 (FAD), B6 (pyridoxal phosphate) and serum B9 (folates) concentrations were in normal range at the beginning of the hemodialysis session in all patients and significantly decreased by 20.2, 13.0, 25.4 and 32.6%, respectively. Mean whole blood vitamin C (ascorbic acid) concentration was in normal range and significantly decreased by 66.6%. Mean whole blood selenium was in normal range and significantly decreased by 6.7%.

By contrast, mean whole blood vitamin B12 (cobalamin) and zinc concentrations, which were both in normal range at the start of the on–line HDF session, were not significantly modified (−2.3 and +1.0%, respectively).

### 3.3. Dialysate Concentration of Vitamins and Trace Elements

During the hemodialysis session, 1% of the total dialysate volume was collected to measure the concentration of the different nutrients. Results are expressed in Table 3 in concentration and absolute mass transfer through the dialysate. We also compared the session absolute loss of nutrients with the RDA in healthy adults.

Important dialysate losses were evidenced corresponding to 97% of RDA for thiamine, 77% of RDA for folic acid, 179% of RDA for vitamin C and 234% of RDA for Zinc. For riboflavin and pyridoxal, dialysate loss was moderate, being 23 and 22% of RDA respectively. Finally, we could not detect any selenium or cobalamin in the dialysate.

## 4. Discussion

Water–soluble vitamins and trace element losses during hemodialysis is an ongoing subject of studies since the first paper published in the 1980s [1]. However, most of these studies involved patients undergoing chronic conventional hemodialysis. Since then, new convection techniques such as post-dilution on-line hemodiafiltration have been progressively implemented in clinical practice in Europe, leading to an improvement of dialyzed patient′s quality of life and life expectancy. However, due to an accrual blood epuration, this new technique may induce uncertain compounds handling and losses.

The present study is the first to show important losses in some vitamins and trace elements during an HDF session. Our population received an adequate dialysis dose in accordance with the latest recommendations [12,13], with a mean Kt/V > 1.2–1.4 and a mean convective volume of 25.5 L per session.

Vitamin C is an important antioxidant molecule and is a cofactor of several enzymes in humans (involved in norepinephrine and collagen synthesis for example). It was lost by a large magnitude of 147 mg per session in the dialysate, corresponding to almost twice the RDA. This result is consistent with the available literature. Morena et al. [14] investigated the convective and diffusive losses of vitamin C during hemodiafiltration and found a mean loss of 66 mg/session (8–230 mg), one third of which was attributed to the convective component alone. Using a conventional low–flux hemodialysis, Chazot et al. [15] reported a mean loss of 80–280 mg per session. Thus, hemodiafiltration may further increase vitamin C losses. It is important to note that 40% of our patients had pre-dialysis plasma values below normal despite a systematic oral supplementation implemented in our center. This trend might be accentuated by the low food intake of vitamin C [16] from the low potassium diet generally prescribed to these patients. Therefore, systematic vitamin C plasma monitoring and supplementation are paramount for patients undergoing hemodialysis and particularly during hemodiafiltration.

Vitamin B1 is best known for the cofactor role of its phosphorylated derivates in enzymatic complexes, participating in order to maintain a normal cellular metabolic function. Vitamin B1 indeed exists on various form in human blood: un-phosphorylated form as thiamine and phosphorylated form as thiamine pyrophosphate. Thiamine pyrophosphate (TPP) is the most abundant biologically active form in the human body and represents 80% of the total thiamine content. Monitoring TPP in whole blood is classically carried out to access vitamin B1 status. For vitamin B1, the loss in dialysate was moderate, corresponding to 1.12 mg, which is roughly equivalent to the RDA and comparable to the results of Schwotzer et al. [17]. Because of a lack of sensibility of our technique, thiamine pyrophosphate was undetectable in the dialysate for most patients, and we only measured the inactive form of vitamin B1 (thiamine). There are conflicting data available on the loss of thiamine during HD, as different dosing methods were used [2,18,19]. Moreover, low thiamine pyrophosphate blood level in MHD patients is infrequent at the beginning of the session [20,21], a finding that is confirmed in the present study, probably in response to the western diet and the absence of loss into the dialysate. However, because of the functional thiamin deficiency in plasma that is well described in end–stage renal disease [2,22], dietary vitamin intake must be evaluated and supplemented if necessary.

Riboflavin, FMN and FAD, all vitamers of vitamin B2, are involved in numerous biological processes such as carbohydrate, nucleic acid, and amino acid metabolism. Only an inactive form of vitamin B2 was detected in the dialysate, corresponding to a moderate loss of 0.28 mg or 23% of RDA. Active forms of vitamin B2 could not be detected, as they only exist in the intracellular space. All patients had normal pre–dialysis flavine adenine dinucleotide values, even if they did not receive any systematic supplementation in our center. Descombes et al. found the same results in a study of conventional hemodialysis [2]. On the other hand, in our study (Figure 2) and that of Schwotzer et al. [17], blood concentrations slightly decrease during online HDF session, which can be indicative of vitamin B2 losses. Moreover, Jankowska et al. [23] found that 48% of MHD patients had inadequate vitamin B2 intake. Consequently, for patients undergoing maintenance HDF, dietary vitamin B2 intake should be assessed, but supplementation may not be warranted.

Vitamin B6 is essential to neo-glucogenesis and is involved in the functioning of the central nervous system, especially in neurotransmitter synthesis. For this micronutrient, only pyridoxal (that is a vitamin B6 unphosphated form) was present in the dialysate and the total loss in one session corresponded to 22% of the RDA (0.33 mg). Recently, Schwotzer et al. [17] found a lesser amount of vitamin B6 loss in the dialysate in maintenance HDF patients, corresponding to about half of the dialysate loss we found in the present study. However, in their study, different dialysis membranes were used and convective volumes were slightly lower. Vitamin B6 deficiency is common in MHD patients with no systematic supplementation [24,25] and up to 50% deficient patients based on pre–dialysis levels. Moreover, 40% of MHD patients had low dietary vitamin B6 intakes in a recent study [23]. Therefore, as previously suggested [26], a systematic supplementation of vitamin B6 is relevant in the era of online HDF.

Folates are required for purine synthesis and are crucial for the re-methylation of homocysteine. Hyper-homo-cysteinemia is considered as an independent biomarker of cardio–vascular risk [27]. In the present study, vitamin B9 losses were important, with a mean per session loss of 0.31 mg into the dialysate corresponding to 77% of RDA, along with a 33% plasma reduction (Figure 4). These data are similar to the study of Schwotzer et al. [17] who described a loss in dialysate that corresponded to 67% of daily-recommended dose. It also concurs with the study of Leblanc et al. [28] which found a 26% serum folate decrease during one HD session using high–flux dialyzers. Despite these results, the existence of folate deficiency in MHD patients is controversial, and the effect of convective dialysis techniques compared to standard HD remains uncertain [29]. Our results suggest, however, that folate losses are present to a greater extent during HDF compared to standard HD. Chazot et al. [26] recommend a folate supplementation of 0.4–1.0 mg per day. As this supplementation is safe, we suggest following this recommendation during on-line HDF treatment.

Vitamin B12 plays a role in cell division, contributing to normal formation of red blood cells as well as in the functioning of the immune system. We could not detect any evidence of B12 in the dialysate measurements. This could be explained by the lack of sensibility of the assay technique but also by the large molecular weight of cobalamin and its carrier proteins in plasma that could prevent diffusion through the dialyzer membrane. Cobalamin losses are probably low in hemodiafiltration, a hypothesis confirmed by the available literature [30]. We found only one study showing presence of vitamin B12 in dialysate [31], but the accuracy of the assay used was uncertain according to the authors themselves. This study also showed no adsorption by the dialyzer membrane. Therefore, no systematic supplementation of vitamin B12 seems necessary in chronic HDF patients but regular monitoring of B12 blood levels might still be relevant in this population.

Zinc plays an important role in immune response, hemoglobin metabolism and oxidative stress. The results of the present study on zinc losses during HDF are contradictory. While we found no decrease in plasma zinc concentration during the HDF session, the amount found in the dialysate was important, with a mean loss of 25.75 mg per session, corresponding to 234% of RDA. We did not find other reports in which zinc losses into dialysate were directly measured. Older studies also found no decrease in zinc concentration during one standard hemodialysis session [32]. This could be explained by a dynamic regulation of zinc from major reservoirs of intracellular zinc and rapid blood release, allowing constant blood levels even despite the HDF clearance [33,34]. Because of the important losses into the dialysate highlighted in the present study, and the fact that 40% of MHD patients were zinc deficient in a previous report by Bozalioglu et al. [35], a systematic supplementation of this trace element should be proposed in chronic HDF patients.

Selenium is an important trace element that also plays a key role in immune function and control of oxidative stress. Since the 1980s, its role in the prevention of cancer [36,37] and cardiovascular death [38] is also well established. In MHD patients, selenium levels are decreased compared to healthy controls [39], and this may induce development of atherosclerosis and coronary disease [40]. Bogye et al. [41] have shown selenium losses during one HD session using poly-sulfone dialyzer membrane that were closely correlated to the protein losses into the dialysate. Contrary to this study, we could not detect any selenium in the dialysate, while selenium concentrations slightly but significantly decreased by 6.7% between the beginning and end of the HDF session (Figure 8). Due to its important role in cancer pathogenesis and cardiovascular events, selenium concentration requires close monitoring and an adequate supplementation should be made when necessary.

## 5. Conclusions

This study is the first to extensively evaluate the loss of water-soluble vitamins and trace elements during one on-line hemodiafiltration session in a significant number of maintenance HD patients. We found important losses into the dialysate, especially for vitamin B1, B6, B9, C and zinc, prompting monitoring of blood levels and systematic supplementation of these compounds in HDF patients.

## Figures and Tables

**Figure 1 nutrients-14-03454-f001:**
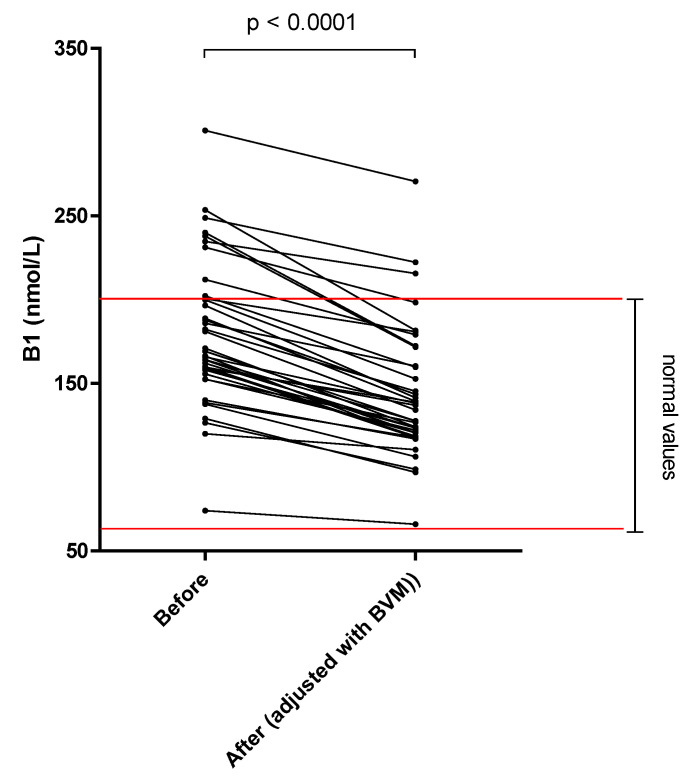
Vitamin B1 (thiamine pyrophosphate) blood concentration before and after one 4 h-HDF session (paired *t*-test).

**Figure 2 nutrients-14-03454-f002:**
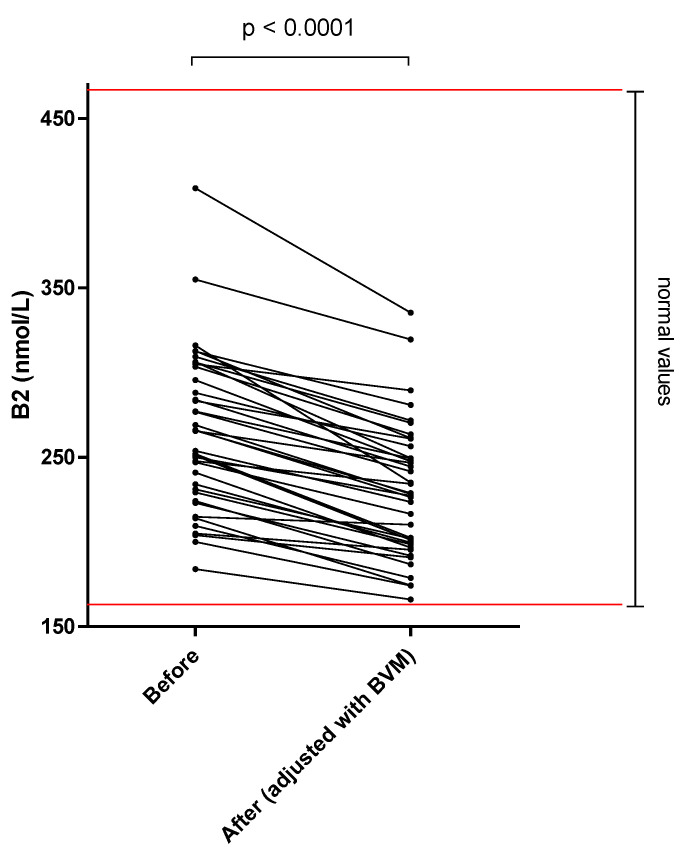
Vitamin B2 (FAD) blood concentration before and after one 4 h-HDF session (paired *t*-test).

**Figure 3 nutrients-14-03454-f003:**
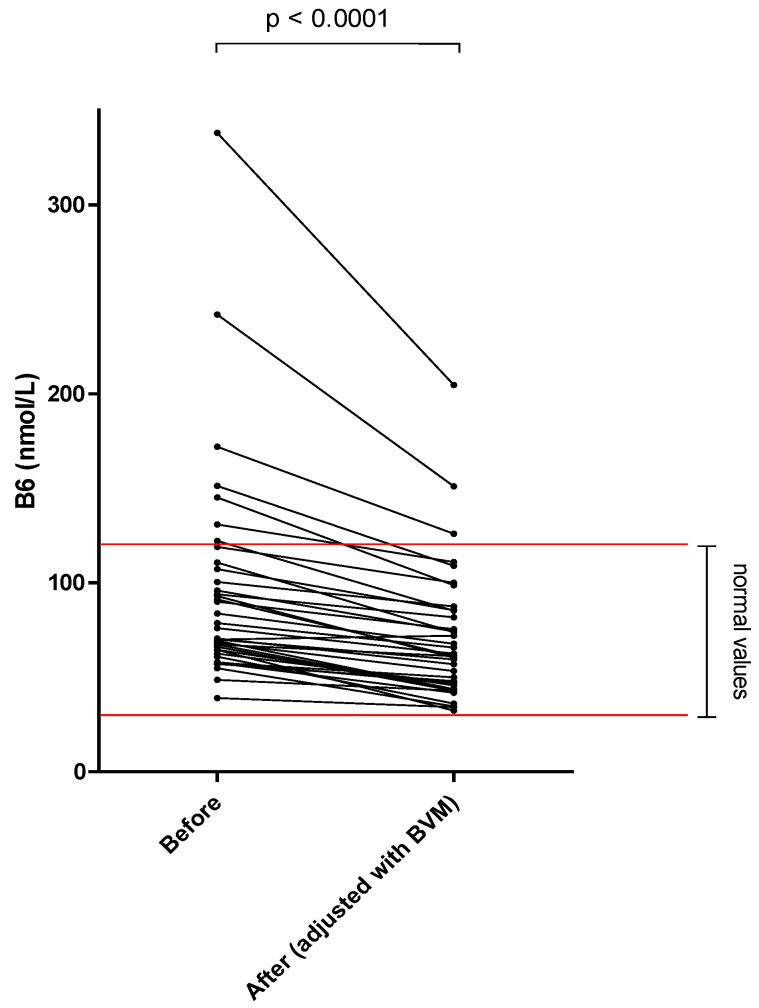
Vitamin B6 (pyridoxal phosphate) blood concentration before and after one 4 h-HDF session (paired *t*-test).

**Figure 4 nutrients-14-03454-f004:**
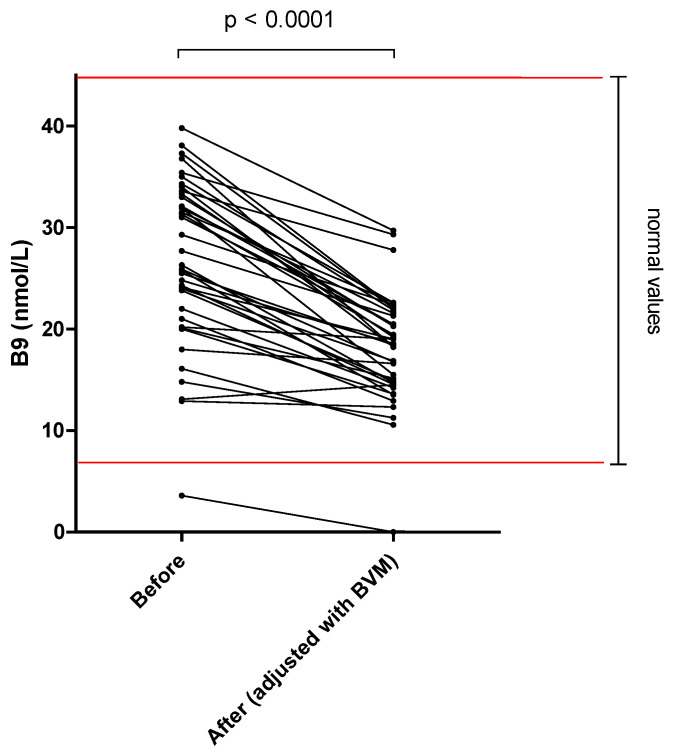
Vitamin B9 (folates) serum concentration before and after one 4 h-HDF session (paired *t*-test).

**Figure 5 nutrients-14-03454-f005:**
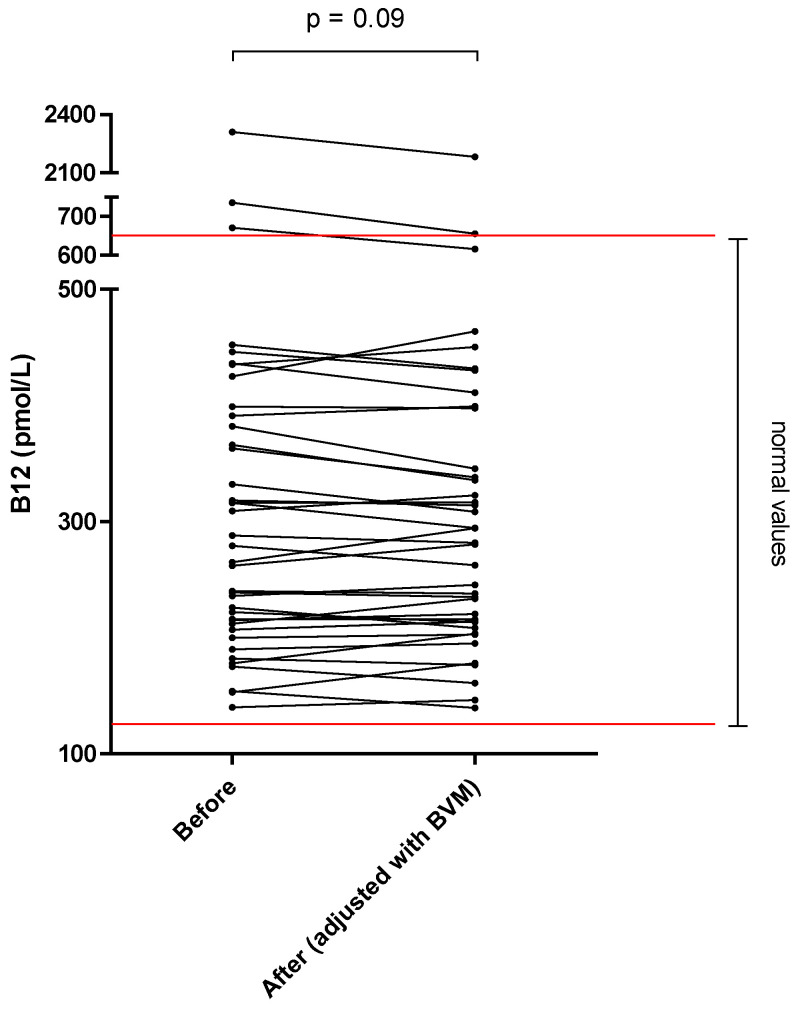
Vitamin B12 (cobalamin) blood concentration before and after one 4 h-HDF session (paired *t*-test).

**Figure 6 nutrients-14-03454-f006:**
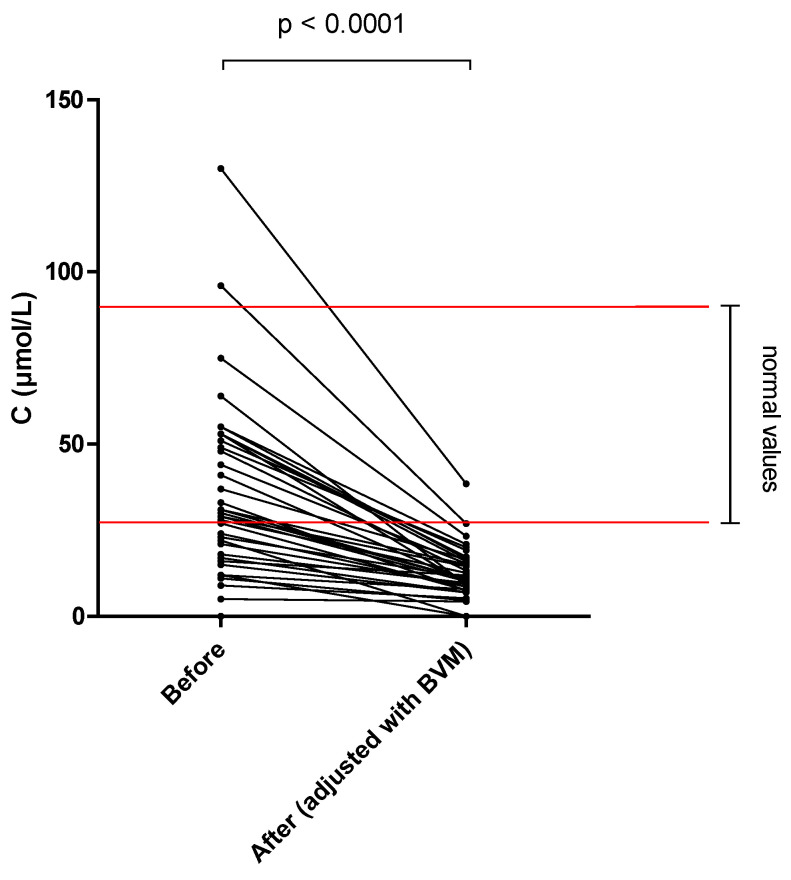
Vitamin C (ascorbic acid) blood concentration before and after one 4 h-HDF session (paired *t*-test).

**Figure 7 nutrients-14-03454-f007:**
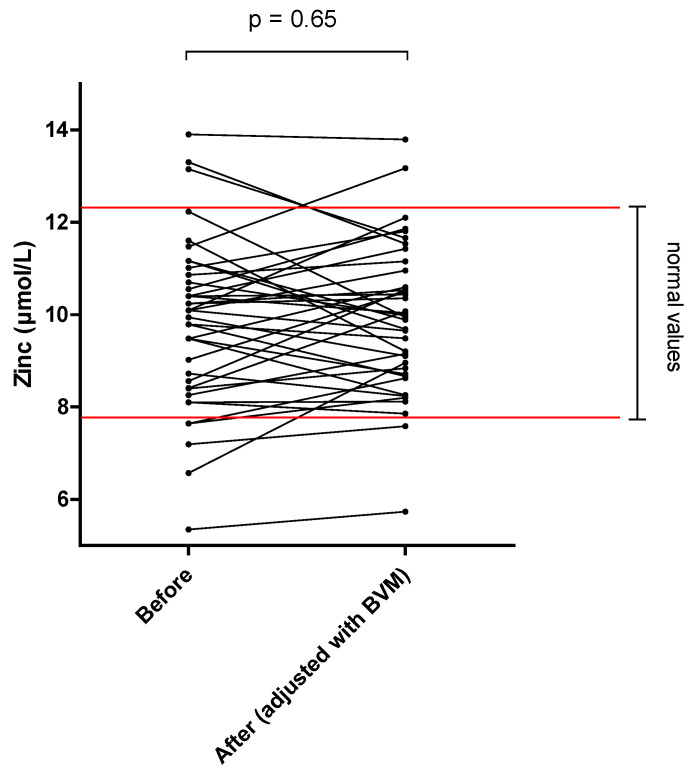
Zinc blood concentration before and after one 4 h-HDF session (paired *t*-test).

**Figure 8 nutrients-14-03454-f008:**
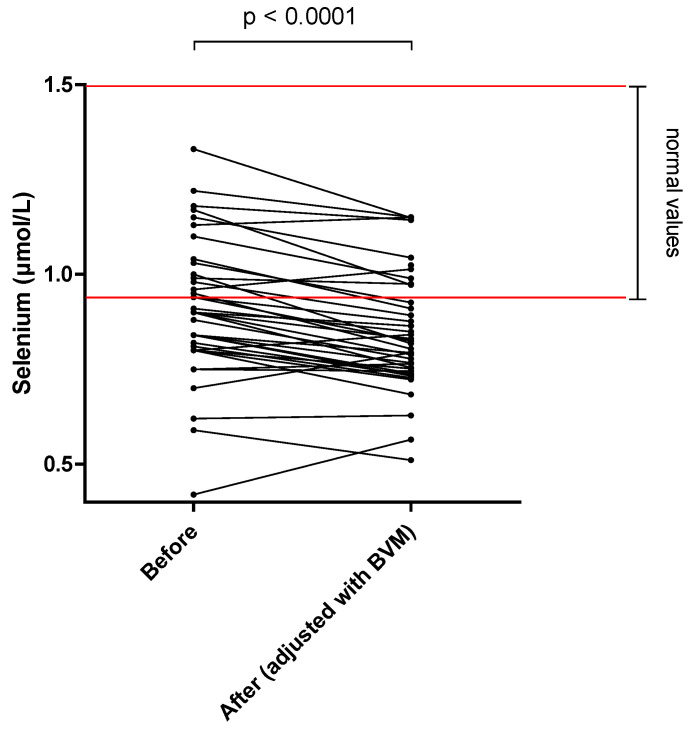
Selenium (thiamine) blood concentration before and after one 4 h-HDF session (paired *t*-test).

**Table 1 nutrients-14-03454-t001:** Baseline characteristics of hemodialysis patients (N = 39).

	Mean ± SD or n
Gender, female	10
Age, year	67.0 ± 14.8
Weight, kg	76.6 ± 14.1
Height, m	1.67 ± 0.08
BMI, kg/m^2^	27.6 ± 5.7
Diabetes (Yes/No)	15/24
Vascular access (AVF/Catheter)	35/4
Dialysis vintage (months)	44 ± 33
Dialysis session duration (min)	234 ± 4
Blood flow (mL/min)	372 ± 25
Kt/V(ocm) ^ǂ^	1.56 ± 0.23
Convective volume per session (L)	25.5 ± 2.5
Effluent collected per session (L)	1.6 ± 0.3
Total Effluent volume per session (L)	166.2 ± 29.6
Relative Blood Volume at the end (%)	88.3 ± 4.6
Serum Albumin (g/L)	38.2 ± 3.8
Serum Pre–Albumin (g/L)	0.33 ± 0.08
Hemoglobin (g/dL)	11.0 ± 0.9

BMI: body mass index; AVF: Arterio-veinous fistula; ǂ ocm: online clearance monitoring.

**Table 2 nutrients-14-03454-t002:** Vitamins and trace elements blood concentration before and after one hemodialysis session.

	Normal Range	Pre Session Concentration	Post Session Concentration	% Change ^¶^
Thiamine pyrophosphate ^Δ^ (nmol/L)	66–200	178.7 ± 43.3	142.7 ± 38.6	–20.2 *
FAD ^Ε^ (nmol/L)	174–471	264.6 ± 46.4	230.2 ± 39.4	–13.0 *
PLP ^Φ^ (nmol/L)	35–110	94.4 ± 56.0	70.4 ± 35.2	–25.4 *
Folates ^Γ^ (nmol/L)	7–46	26.7 ± 8.1	18.0 ± 5.4	–32.6 *
Cobalamin ^H^ (pmol/L)	138–652	355.9 ± 346.8	347.6 ± 323.9	–2.3
Ascorbic acid ^Ι^ (μmol/L)	25–85	34.5 ± 26.6	11.5 ± 7.9	–66.6 *
Zinc (μmol/L)	7.8–12.4	9.8 ± 1.8	9.9 ± 1.6	+1.0
Selenium (μmol/L)	0.9–1.5	0.9 ± 0.2	0.85 ± 0.15	–6.7 *

Δ vitamin B1, ^Ε^ flavin adenine dinucleotide or vitamin B2, ^Φ^ Pyridoxal–5′–phosphate or vitamin B6, ^Γ^ vitamin B9, ^H^ vitamin B12, ^Ι^ vitamin C, ^¶^ after volume reduction correction (refer to methods), * *p* < 0.001.

**Table 3 nutrients-14-03454-t003:** Vitamins and trace elements parameters in dialysate.

	Dialysate Concentration	Dialysate Mass Loss	RDA Equivalent
Thiamine	20.71 (±17.92) nmol/L	3321.0 ± 2613.0 nmol 1.12 ± 0.88 mg	97%
Riboflavin	4.61 (±4.97) nmol/L	751.7 ± 804.1 nmol 0.28 ± 0.30 mg	23%
Pyridoxal	8.07 (±2.44) nmol/L	1326.0 ± 385.2 nmol 0.33 ± 0.09 mg	22%
Folates	4.29 (±2.40) nmol/L	715.1 ± 404.6 nmol 0.31 ± 0.18 mg	77%
Cobalamin	ND	ND	
Ascorbic acid	5.13 (±5.00) μmol/L	836.9 ± 825.9 μmol 147.50 ± 145.50 mg	179%
Zinc	2.37 (±0.44) μmol/L	394.4 ± 105.8 μmol 25.75 ± 6.91 mg	234%
Selenium	ND	ND	

RDA: recommended dietary allowance in healthy adults per day; ND: not detectable.

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
