# Peer review of "Water-Soluble Vitamins and Trace Elements Losses during On-Line Hemodiafiltration"

_nutrients, 2022, doi:10.3390/nu14173454_

Round 1

Reviewer 1 Report

Dear Authors:

Regarding the manuscript with title “Water-soluble Vitamins and Trace Elements Losses during On–line Hemodiafiltration”, I have one major comment and some minor comments to address.

Major Comment:               

Lines 285-286: “Our population was representative of the European population undergoing maintenance hemodialysis”. To comprove this sentence, authors have to do sample size calculation and refer its value on Methods

Minor Comments:

Comment 1:

Lines 11-13: Authors must rephrase this sentence: Maintenance hemodialysis induces water-soluble vitamins and trace elements losses and most recommendations regarding potential supplementation are based on conventional hemodialysis

Comment 2:

Lines 18-19: Authors must change “No significant decrease was found for vitamin B12 and zinc” by “No significant differences were found for vitamin B12 and zinc”

Comment 3:

Lines 22-23: In conclusion, during a standard 4hr-HDF session, we found important losses for vitamin B1, B6, B9, C and zinc

Comment 4:

On Keywords, authors must change “Vitamins” by “Water-soluble vitamins”

Comment 5:

On the first sentence of Introduction, authors must add a sentence related to the definition of Hemodialysis to contextualize the readers.

Comment 6:

Line 28: Authors must change “Maintenance hemodialysis (MHD) patients” by “On Maintenance hemodialysis (MHD), patients”

Comment 7:

Lines 29-31: “The origin of these deficiencies is not fully understood and involve multiple mechanisms such as decreased food intake, gastrointestinal malabsorption, altered vitamin metabolism and loss during the hemodialysis session itself”. Authors must add at least one reference regarding the previous sentence.

Comment 8:

Line 35: Authors must change “Among MHD” by “Among MHD,”

Comment 9:

Line 51: “from our hemodialysis center” Authors must clarify the location of the hemodialysis center”

Comment 10:

On chapter 2.1. authors must refer that this is a prospective study

Comment 11:

Line 69: Authors must change ‘arterial line’ line by ‘arterial line’

Comment 12:

Line 173: Authors must change “Levels of Se” by “Levels of Selenium (Se)”

Line 176: Authors must change “Levels of Zn” by “Levels of Zinc (Zn)”

Comment 13:

Line 289: Authors must insert a sentence referring the importance of vitamin C

Author Response

Dear reviewer, 

Thank you very much for your review and your help with our article. 

Please see the attachment with correction. 

Kind regards

Reviewer 2 Report

The manuscript reported  the measurement of soluble vitamins (B1, B2, B6, B9, B12, C) and trace elements (Zn and Se) losses during one on–line post–dilution hemodiafiltration (HDF) session of thirty-nine patients. Blood and dialysate samples were analysed before and after one HDF showing significantly decreased for B1 B2, B6 , B9 , C and selenium, suggesting the importance of regular monitoring of plasma levels and systematic supplementation of these compounds. The article is clearly, well written and of great interest.

Author Response

Dear reviewer, 

Thank you very much for your comment. 

Kind regards